# “I Don’t Want to Go to Work”: A Mixed-Methods Analysis of Healthcare Worker Experiences from the Front- and Side-Lines of COVID-19

**DOI:** 10.3390/ijerph20115953

**Published:** 2023-05-25

**Authors:** Smith F. Heavner, Mackenzie Stuenkel, Rebecca Russ Sellers, Rhiannon McCallus, Kendall D. Dean, Chloe Wilson, Marissa Shuffler, Thomas W. Britt, Shannon Stark Taylor, Molly Benedum, Niki Munk, Rachel Mayo, Kathleen Buford Cartmell, Sarah Griffin, Ann Blair Kennedy

**Affiliations:** 1Department of Public Health Sciences, Clemson University, Clemson, SC 29634, USA; 2Department of Biomedical Sciences, University of South Carolina School of Medicine Greenville, Greenville, SC 29605, USA; 3Premier Neuro, Greer, SC 29650, USA; 4Department of Psychology, Clemson University, Clemson, SC 29634, USA; 5Center for Family Medicine, Department of Medicine, Prisma Health, Greenville, SC 29605, USA; 6AppFamily Medicine, Department of Medicine, Appalachian Regional Healthcare System, Boone, NC 28607, USA; 7Department of Health Sciences, School of Health & Human Sciences, Indiana University, Indianapolis, IN 46202, USA

**Keywords:** burnout, moral distress, healthcare worker, nurse, massage therapist

## Abstract

During the COVID-19 pandemic, healthcare workers (HCW) were categorized as “essential” and “non-essential”, creating a division where some were “locked-in” a system with little ability to prepare for or control the oncoming crisis. Others were “locked-out” regardless of whether their skills might be useful. The purpose of this study was to systematically gather data over the course of the COVID-19 pandemic from HCW through an interprofessional lens to examine experiences of locked-out HCW. This convergent parallel mixed-methods study captured perspectives representing nearly two dozen professions through a survey, administered via social media, and video blogs. Analysis included logistic regression models of differences in outcome measures by professional category and Rapid Identification of Themes from Audio recordings (RITA) of video blogs. We collected 1299 baseline responses from 15 April 2020 to 16 March 2021. Of those responses, 12.1% reported no signs of burnout, while 21.9% reported four or more signs. Qualitative analysis identified four themes: (1) professional identity, (2) intrinsic stressors, (3) extrinsic factors, and (4) coping strategies. There are some differences in the experiences of locked-in and locked-out HCW. This did not always lead to differing reports of moral distress and burnout, and both groups struggled to cope with the realities of the pandemic.

## 1. Introduction

Pandemics, such as COVID-19, impact the mental health of healthcare workers (HCW) and are specifically associated with increased levels of stress, burnout, and moral distress [1,2,3]. HCW treating infected patients are found to have increased fears of disease transmission to family and self-infection [4]. The COVID-19 pandemic exposed ill-equipped health systems and resource scarcity, resulting in increased rates of moral injury for HCW tasked with challenging ethical decision-making [5]. Coupled with the impact of pandemic-related stress and increased rates of fear, anger, depression, and anxiety in the general population, HCW are at increased risk for mental health crises [6,7]. Ubiquitous disruption, continuous exposure to high-intensity stress, and a staggering death toll during the COVID-19 pandemic highlight an urgent need to prioritize the emotional and physical needs of HCW [8,9]. Additionally, systems and organizational pressures seem to mostly contribute to the burnout of healthcare providers rather than provide support to staff [10].

Research into the burden of the COVID-19 pandemic on HCW is sometimes limited to single perspective/site narratives, and fails to examine the emotional, mental, and financial burden of HCW who are locked-out of the healthcare system due to policy, location, or profession, despite a desire to help. Significant research explores the experience of nurses and physicians, especially those working in critical care units [11]. A growing body of literature examines and compares the COVID-19 experiences of professional groups such as clinicians and academic professionals [12] and social workers [13], but these are again professions which were forced to adapt their workflows and client interactions, not halt them [14].

The purpose of this study was to systematically gather data over the course of the COVID-19 pandemic from HCW through an interprofessional lens to examine provider experiences from two perspectives: (1) HCW who are locked-in the healthcare system with little control to make changes (e.g., physicians, nurses, respiratory therapists), and (2) HCW locked-out of healthcare systems (e.g., dentists, massage therapists, mental health practitioners).

Three research questions drive this study: (RQ1) How do locked-in and locked-out HCW feel prepared and supported during the COVID-19 pandemic? (RQ2) To what extent do locked-in and locked-out HCW report feelings of well-being, burnout, or moral distress? Finally, (RQ3) How do locked-in and locked-out HCW describe their experiences of the COVID-19 pandemic? With a potential impact on public health policy and practice, this study presents preliminary data from Project COPE to inform the broadest patterns of HCW experiences to direct health system practices and support of HCW.

## 2. Materials and Methods

Project COPE is an IRB-approved (University of South Carolina) mixed-methods study launched in April 2020 and led by an interdisciplinary team of researchers. HCW from all specialties were recruited to participate using social media platforms, including Facebook™, Instagram™, and LinkedIn™, and other community outreach (e.g., podcasts and newsletters). Interested individuals were directed to a secured data collection site (Qualtrics™) to share their experiences during the pandemic via an anonymous survey. Quantitative and qualitative data were captured in an initial survey and invited to participate in weekly follow-ups. No geographical restrictions were placed on recruitment.

The initial survey screened participants for inclusion criteria (i.e., aged 18 or older, working as a healthcare provider/worker or an active student, able to read/understand English). Demographics, profession, practice setting, years of experience, COVID-19 impacts on practice, and a validated measure of moral distress [15] were included in the initial survey. Participants were invited to submit a 5-min video blog (vlog) in response to one or more of the following prompts: (1) What types of experiences led you to feel moral stress this week? (2) How are you feeling about the next time you go to work? or (3) How are you coping with the pandemic while you are at home? Finally, participants were invited to opt into weekly follow-up surveys by providing an email address.

Participants electing to participate in the weekly surveys received a personalized survey invitation link via email, allowing the researchers to track measures longitudinally. This survey repeated the initial survey moral distress measure and included the Well-Being Index (WBI) [16] and a single-item burnout measure, the “Mini-Z” [17]. Additional questions included whether respondents saw patients that week, if any family members or friends had been infected with COVID-19, potential stressors (perceived lack of personal protective equipment [4], insufficient communication [2], fears for personal and family safety, and stigmatization [18]), and respondents’ coping strategies. Specifically, coping was assessed by a single item from the prompt “Which of the following have you found helpful in the past week?” Responses included a mix of 20 positive, evidence-based strategies, such as seeking support from coworkers, exercise, or recognizing meaning in their work, and maladaptive strategies, such as alcohol or tobacco use [19,20,21,22].

All survey responses were analyzed following a convergent parallel mixed-methods design. Researchers concurrently implemented the quantitative and qualitative portions of the study. The qualitative data was prioritized over the quantitative, yet kept separate until each portion of the analysis was completed [23,24,25]. While merging qualitative and quantitative data, researchers considered how each data stream contextualized the other, and what meaning might be gleaned from observing confirmation, disconfirmation, or expansion of one data source with and from the other [23]. This process is depicted in Figure 1.

Qualitative analysis of 257 video blogs followed the Rapid Identification of Themes from Audio Recordings (RITA) methodology where recordings are analyzed in one-minute segments [26]. Following a deductive/inductive thematic analysis, two evaluators created a detailed codebook of themes, including themes deduced from the literature, and induced from vlogs collected in Project COPE. Inductive theme development was conducted collaboratively, with evaluators viewing video segments together and comparing notes and coding in real-time.

Quantitative analysis was limited to the baseline and first weekly survey collected from each respondent due to a skew in long-term participation towards complementary and integrative HCW (i.e., massage therapists). Moreover, to correct a significant sampling bias towards individuals identifying as white women, we collapsed race and gender into binary categories of “other races and ethnicities versus white non-Hispanic/non-Latinx” and “other gender identities versus women”. Burnout, moral distress, and well-being scores were modeled as ordinal scales and collapsed into binary “high” and “low” categories. Four or more items on the WBI, a score of 4 or 5 on moral distress [16], and a score of 4 or 5 on the Mini-Z were considered “high” [17]. Professional categories were created to aid in analysis. These categories included allied health (surgical technicians, radiology technicians, athletic trainers, occupational therapists, and physical therapists), medicine (physicians, advanced practice providers, respiratory therapists, emergency medical technicians, and paramedics), mental health (psychiatrists, counselors, and social workers), and other (acupuncturists, chiropractors, dentists and dental technicians, midwives, naturopaths, optometrists, and pharmacists). Due to the large numbers in each group, nurses and massage therapists were left in their own categories.

Quantitative analysis included a series of logistic regression models. Model specification was informed by extant literature but was limited by overrepresentation of some professions (i.e., massage therapy). Burnout, moral distress, and well-being scores in HCW were assessed in three separate cumulative logistic regression models, controlling for demographics and years of experience, and stratified by thematic clustering of professions. For both variables, analyses considered the “other” category as the reference group. In addition to modeling the full ordinal scales, measures were also collapsed into binary “high” and “low” categories within the logistic regression models.

To further compare professionals across thematic clustering, differences in coping strategies and work concerns were explored between those seeing patients and those not seeing patients using single-tailed Fisher’s exact tests. Statistical analyses were performed using Microsoft Excel™ (Microsoft Corporation. Redmond, WA. 2018. Retrieved from https://office.microsoft.com/excel) and SAS™ software version 9.4 (SAS Institute. Cary, NC, USA). Alpha was set at 0.05 for all analyses.

## 3. Results

This study included 1299 baseline participant responses from 15 April 2020 to 16 March 2021. Project COPE respondents overwhelmingly identified as women (83.7%) and white (80.6%). Of the baseline survey respondents, 70.7% (918) enrolled in weekly follow-up surveys and provided a valid email address. These participants submitted a total of 2411 weekly responses and 257 vlogs over the study period. Demographics on the full study sample at baseline are given in Table 1.

Results are presented under four themes and six subthemes. Themes include: (1) professional identity (a desire to help versus a feeling of being stuck or trapped), (2) intrinsic stressors (e.g., fear of COVID-19, allostatic load), (3) extrinsic factors (policy and financial impact), and (4) coping strategies. Quantitative results are nested under themes to demonstrate how mixed-method results informed interpretations and theme construction. Qualitative notes on appearance, demeanor, and context are included with quotes to ascribe additional depth.

### 3.1. Theme 1: Professional Identity

At baseline, this sample included, among others, 592 (45.6%) massage therapy professionals and 273 (21.0%) nursing professionals. A small contingent of 81 (6.2%) respondents identified as dual-role professionals (e.g., holding both nursing and massage therapy credentials). Respondents almost universally framed their response and experience of the pandemic in their professional identity. Two subthemes emerged: a desire to help (locked-out) and a feeling of being stuck or trapped (locked-in).

Multiple respondents who identified as massage therapists discussed cross-training or holding additional certifications (e.g., registered nurse or lab technician) during vlog submissions, but only indicated massage therapy as their profession on intake surveys. This included one massage therapist who described her experience as a new registered nurse working on a COVID unit. She sits on the floor in front of a pair of folding doors. Her hair is wet and her face red. As she speaks, she looks away from the camera, and she begins to slouch as she speaks:

I realized I am the one that is in contact every single day that I work with COVID patients, so… I had an epiphany that I will not be able to do massage for a really long time.

Like other respondents, she reflected on the realities of holding dual professions.

#### 3.1.1. Subtheme 1.1: Locked-Out

Many participants, particularly massage therapists, described feeling cut off from their work and sought to find meaning in other ways. One respondent exemplified this experience and explained her decision to stop seeing clients, citing the risk of spreading COVID-19. The video is well lit, and she sits in front of an overflowing bookcase. Her hair is fixed with stiff curls, and her eyes are bright, but the camera shakes as she speaks:

I don’t know what it’s going to feel like the next time I have my hands on someone. I think it’s going to feel really great, but I also think it’s going to feel really scary.

In the baseline survey, a total of 768 respondents reported that they stopped seeing patients. Participants described this interruption in patient care as required by government action (403, 52.5%), employer (132, 17.2%), professional association (66, 8.6%), educational institution (81, 10.6%), recommended by employer (28, 3.6%) or professional association (181, 23.6%). Eighty-one (10.6%) respondents reported being furloughed or laid off, and 362 (47.1%) described the decision to stop seeing patients as a personal choice.

#### 3.1.2. Subtheme 1.2: Locked-In

Other HCW, particularly those working in acute care settings, described a sense of feeling trapped. In one video blog, there are dark circles under the respondent’s eyes and a faint redness on the bridge of their nose and down their cheeks, presumably from wearing a respirator at work. Their hair is pulled into a messy ponytail, and they are wearing a wrinkled, grey t-shirt.

I’ve even thought about calling in sick for shifts because I just don’t want to go in and have to deal with all of this stuff… I don’t want to go to work.

At baseline, 521 respondents reported they were continuing to see patients. Changes in patient interactions included incorporating telehealth (137, 26.3%), decreased patient load (230, 44.2%), increased patient load (195, 37.4%), assisting in critical care (99, 19.0%), and assisting in other departments (127, 24.4%).

Many respondents described feeling unsupported and ill-equipped to provide care to COVID-19 patients. One advanced practice provider said in a video blog:

Yes, I chose to do training in medicine, and yes I wanted to be there, but I didn’t want to feel like the resources weren’t there or feel like I couldn’t provide the right answers.

### 3.2. Theme 2: Intrinsic Stressors

The second theme centers on stress and anxiety related to the pandemic and other social factors. Participants expressed a sense of responsibility to help control these stressors. Two subthemes were identified: (1) Fear of COVID-19 and concern about spreading Covid, and (2) “Allostatic Load”.

#### 3.2.1. Subtheme 2.1: Fear of COVID-19 and Concern about Spread

Fear of Covid included general concerns about death and suffering, but, more often, many respondents specifically highlighted aspects of the disease process they feared experiencing or watching others experience. Concerns over the spread of COVID-19 took two distinct forms: concern about spread in the clinical environment, and a fear of spreading COVID-19 to family members and friends. This led to feelings of isolation as respondents struggled to balance anxieties and reasonable precautions. In an extreme example, a medical assistant and massage therapist reported testing positive. She is visibly short of breath:

Unfortunately, because I felt fine on Saturday, I saw three of my massage clients, and so they were all exposed. I was wearing a mask and a face shield… [but] one of the three have tested positive.

Her face falls, and she sits in silence for a few seconds.

Concern about spreading the virus was even prevalent among students. A first-year medical student described her concerns visiting her parents:

There was quite a bit of stress in that having parents above the age of 65 and have comorbidities and then coming to a family’s house and having to deal with potentially spreading the virus to them unknowingly. It’s the unknowing part of COVID that’s the most stressful.

These concerns were also represented in our quantitative results. Of the respondents, 71.4% endorsed fears of loved ones getting sick or dying, 52.1% endorsed concern for their own health, and 35.9% indicated concerns about their colleagues falling ill. A higher proportion of those seeing patients reported feeling isolated or lonely (52.1% vs. 37.5%, *p* = 0.002) and concerns about care of family members (16.7% vs. 9.9%, *p* = 0.04). Respondents who reported not seeing patients endorsed concerns about personal resources, such as financial concerns, advanced directives, and home supplies (59.4% vs. 43.2%, *p* = 0.001). Full comparison of personal concerns between locked-in and locked-out HCW is included in Table 2.

#### 3.2.2. Subtheme 2.2: Allostatic Load

Some respondents felt the burden of additional social, political, and cultural stress added to concerns about COVID-19, including the contention surrounding the 2020 US presidential election and racial disparities highlighted by the death of George Floyd. Participants pointed out the pandemic was not occurring in a vacuum. Comments about politics, the Black Lives Matter movement, and the environment were made exclusively by those with “non-essential” roles, but some did hold dual certification. For example, a massage therapist shared her excitement about finishing a medical assistant course and getting to start work in an urgent care unit; then she shifted to describing struggling with the political views of her massage clients:

I’m worried that if I make [wearing masks] a mandatory thing, I’m going to lose that client…It’s a very conservative town, and a lot of people don’t like to wear masks.

She is pacing through her house, presumably recording on her phone which is held low in her hand.

One respondent, who did not disclose their profession, stated in a video blog:

I have a lot of people that are stressed by the politics and the election…There’s different stress stimulus that wasn’t present a few months ago.

She sits close to the camera, and the light from the screen highlights heavy circles under her eyes.

### 3.3. Theme 3: Extrinsic Factors

Respondents frequently discussed economic impacts of the pandemic including being furloughed or laid off, as well as making decisions to open or close their businesses or to furlough or lay off team members. This was frequently seen among locked-out HCW, but one nurse described her experience:

Our hospitals have a huge decrease of patients, which means starting next week we start furloughs…which is a very strange feeling, thinking that you’re very important and you’re going to do all this good, [then] you are not needed.

One respondent described frustrations of her practice and work as a massage therapist being labeled as “non-essential” by her state. She sits on the floor in her bathroom, leaning against the cabinets wearing a bathrobe. She describes her decision to comply with the state orders despite hearing of other practitioners who have found loopholes to continue practicing, such as being part of a medical office.

I’d be putting my livelihood, my business, and my professional recognition at stake if I were to justify to myself that medically necessary massage was ok with the mandates that our governor has made.

Descriptive statistics of work-related experiences of locked-in HCW (Table A1 and Table A2) and a comparison of work-related concerns in locked-in versus locked-out HCW (Table A3) captured by Project COPE are included in Appendix A.

### 3.4. Theme 4: Coping Strategies

Respondents often reported activities intended to help them cope with the realities of the pandemic. Common activities included walking, gardening, reading, and meditation. Several participants described their coping strategies (e.g., smoking, overeating) as “unhealthy,” but often presented activities without judgment. One massage therapist said:

I am sleeping a lot. I don’t get out of bed very often. I have been looking for something to fill my time, so I’ve taken up some gardening and getting some ideas ready for yard work.

Further representing this theme, one nurse shared:

I’ve been coping by eating a lot of comfort food… I’m gaining weight like crazy, like 20 pounds since quarantine started.

She is standing in front of a bathtub. Her hair is fixed with large curls, and she wears a fleece jacket.

Respondents often reported difficulty with feelings of isolation.

I’ve noticed with the downtime the social isolation has gotten to me. Some of my less healthy coping patterns have resurfaced like, um, tobacco smoking and comfort foods, um, and staying up way too late.

Her speech is pressured, and she looks down while speaking, rarely making eye contact with the camera.

In some cases, respondents described activities we coded as “hypervigilance”. This included obsessive news and media consumption, attempts to calculate their own exposure risk, and robust decontamination procedures. Below is an example of this kind of response:

We have a quarantine station we set up in the front door… We wipe everything down as soon as we get home, strip our clothes, they go right into the wash, we go right to the shower before we do anything else when we get home.

The respondent is sitting on her couch. Her hair is wet and her face red. Lines on her face trace the impression of a face mask or respirator.

In another example, one participant submitted an audio-only recording. She speaks slowly, stumbling over her words.

I feel more vulnerable to COVID… I’ve been paying a lot of attention to the number of people I’ve massaged and their occupation. Once I reached a hundred and fifty since we’ve been reopened, I looked at my data and… 11.3% of those 150 are nurses.

A proportion of respondents (n = 111, 27.4%) reported at least one maladaptive behavior including using alcohol (n = 73, 18%), sleeping pills (n = 22, 5.4%), illicit substances (n = 29, 7.2%), or tobacco (n = 14, 3.5%). No significant differences between those who had and had not stopped seeing patients were found. Additional information on coping strategies is included in Table A4 in Appendix B.

### 3.5. Moral Distress and Burnout

A total of 393 participants completed the Well-Being Index (WBI) in the first weekly survey, with 344 (87.5%) reporting at least one symptom of burnout. Forty-nine (12.1%) reported no signs of burnout, while 86 (21.9%) reported four or more. The mean composite score for the WBI was 2.40 (SD = 1.45), indicating low average burnout among participants.

The Mini-Z captured 381 responses in the first weekly survey. The majority of respondents (n = 346 (90.8%)) scored at a level 3 (“I am definitely burning out and have one or more symptoms of burnout”) or lower and 248 (65.1%) scored a 2 (“I am under stress, and don’t always have as much energy as I did, but I don’t feel burned out”) or lower. Only four (1.0%) respondents reported level 5 (“I feel completely burned out.”).

Most respondents reported experiencing moral distress less than once a day, if at all (63.4% and 72.0%, respectively). Those who stopped seeing patients were less likely to report higher levels of moral distress than those who continued seeing patients (OR 0.72; 95% CI 0.52, 0.98) after controlling for years of practice, level of education, and race (dichotomized as non-white versus white). Having stopped seeing patients was also associated with lower odds of reporting higher moral distress in medicine professions (OR 0.31; 95% CI 0.10, 0.96).

In the baseline survey, those who stopped seeing patients were less likely to report higher levels of moral distress than those who stopped seeing patients (OR 0.72; 95% CI 0.52, 0.98). Having stopped seeing patients was also significantly associated with lower odds of reporting higher moral distress in medicine professions (OR 0.31; 95% CI 0.10, 0.96). See Table 3 for additional results.

In the first weekly survey, those who reported they had stopped seeing patients were less likely to report higher levels of moral distress (OR 0.55; 95% CI 0.38, 0.81). Similar trends were observed when outcome measures were treated as binary variables. Those who stopped seeing patients were less likely to report moral distress (OR 0.46; 95% CI 0.28, 0.73). These results are displayed in Table 4.

## 4. Discussion

Through this mixed-methods study, the experiences and perspectives of healthcare providers were explored and stratified across professionals locked-in and locked-out of delivering healthcare services. Several key findings can be drawn from this study: (1) “Locked-in” and locked-out healthcare professionals experienced feelings of burnout, moral distress, and struggled with overall well-being; (2) experiences contributing to these feelings are sometimes varied across professional category; and (3) differences in distinct experiences do not appear to be associated with differences in reported feelings of burnout or moral distress.

Locked-in and locked-out HCW experienced similar concerns about the lack of planning and resources, frustrations with communication and misinformation, and the conflict between professional and personal responsibilities. Differences in concerns related to documentation and organizational policies may be largely due to differences in work settings (e.g., acute care versus outpatient settings) or that locked-out HCW, such as massage therapists, were more likely to be independent practitioners. Further research is needed to illuminate the causes of these patterns.

One example of differences in distinct experiences not being associated with differences in feelings of stress was financial and personal resource concern. Financial and personal resource concern was reported in vlogs of both groups, but locked-out healthcare workers reported concerns more often through vlogs and through Fisher’s Tests of survey responses, stratified by professional category. At least one locked-in professional described concerns about the financial impact of impending furloughs, and 43.2% of locked-in survey respondents reported personal resources as a concern.

This study also found contradicting results in measures of burnout and moral distress, which may be explained by the decisions associated with having a choice on whether to see patients. “Locked-out” professionals were almost half as likely to report higher levels of moral distress, but one and a half times more likely to report higher levels of burnout compared with those who felt “locked-in”. Respondents who indicated they made a personal choice to stop seeing patients were half as likely to report higher levels of burnout on the WBI compared with those who stopped for other reasons. Those who made a choice to stop seeing patients may have seen the decision as a way to contribute to the public health effort to limit disease transmission, and though they may have perceived themselves to be worn down and ranked higher on the Mini-Z, they did not experience the symptoms captured in the WBI. Conversely, those who continued to see patients reported feeling lonely and isolated as well as concerns about dependent care at higher rates than those who had stopped seeing patients. Based on qualitative findings, it may be that those continuing to see patients are more likely to self-isolate out of fear of spreading COVID-19. However, further investigation is recommended.

Both groups endorsed a sense of uncertainty about how long the pandemic would last, fears of social instability, and frustrations about pandemic misinformation. Among locked-out professionals, this was expressed as general anxiety and concern for public health. Project COPE participants who continued to see patients expressed frustrations and fear for their personal well-being and family health. Regardless of profession, COVID-19 was shown to have a prominent impact on healthcare professionals, and responses across both participant groups highlighted the numerous areas of impact both personally and professionally experienced during the pandemic.

## 5. Conclusions

Several limitations were inherent to the study design and were mitigated as much as possible. Through our sampling methods, there was an imbalance in the professional groups included, which limited the rigor of statistical methodologies and our ability to explore the mechanisms of our findings. In addition to professional category distribution, sampling was limited to predominately white women, which may influence the generalizability. Comparison across professional categories was also limited by the applicability of standard measures across healthcare settings. For example, differences in work-related stressors (e.g., bureaucratic tasks, electronic health records) may be due to differences in practice settings. Similarly, phrasing of measures may have influenced respondents’ understanding of the questions or ability to apply it to their perspective. Questions pertaining to organizational support are potentially skewed due to the oversampling of massage therapists, many of whom own their own practices [27].

Project COPE added new voices to the rapidly growing literature on healthcare provider burnout, moral distress, and well-being through longitudinal video recordings with providers both locked-in and locked-out of healthcare services. Similar to other studies, this manuscript highlights the impacts of isolation and fear on healthcare workers but added new evidence around the experiences of professions largely absent from the literature. This study found increased levels of burnout in both categories of providers, but data suggest differing challenges to stress, practice constraints, and disease-related fears. Public health efforts and future research may benefit from studies exploring the distinct mechanisms related to burnout in those locked-in and locked-out of health care services to provide better support to a group that desperately needs it.

## Figures and Tables

**Figure 1 ijerph-20-05953-f001:**
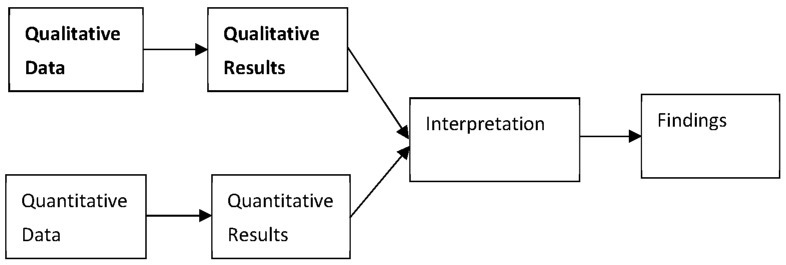
Convergent Parallel Design. Qualitative data and results are bolded to show emphasis.

**Table 1 ijerph-20-05953-t001:** Characteristics of Participants and Study Involvement.

Variable	Baseline Survey N = 1299 (%)	Enrolled in Prospective ^1^	Weekly Responses ^2^
Race			
White, Non-Hispanic/Latinx	1073 (82.6)	762 (71.0)	2402 (84.1)
Other races and ethnicities	226 (17.4)	156 (69.0)	453 (15.9)
Gender			
Women	1118 (86.1)	798 (71.4)	2402 (84.1)
Other Identities	181 (13.9)	120 (66.3)	453 (15.9)
Profession			
Allied Health	54 (4.2)	37 (68.5)	94 (3.9)
Massage Therapy	592 (45.6)	469 (79.2)	1559 (64.5)
Medicine	171 (13.2)	96 (56.1)	214 (8.9)
Mental Health	28 (2.2)	21 (75.0)	43 (1.8)
Nursing	273 (21.0)	178 (65.2)	242 (10.0)
Dual profession	81 (6.2)	56 (69.1)	124 (5.1)
Other	100 (7.7)	61 (61.0)	135 (5.6)

^1^ Percentage is calculated as percentage of each category enrolled in prospective data collection. ^2^ Percentage is calculated as percentage of each category’s contribution to all responses.

**Table 2 ijerph-20-05953-t002:** Personal concerns among locked-in and locked-out HCW.

Which of the Following Are You Currently Concerned about Regarding Yourself? Check All That Apply:	Locked-In N (%)	Locked-Out N (%)	Total N (%)	Χ^2^	*p*-Value
Concern that my colleagues will get sick	77 (40.1)	61 (31.8)	138 (35.9)	2.90	0.06
Fear of getting sick and/or dying myself	105 (54.7)	95 (49.5)	200 (52.1)	1.04	0.18
Fear of my loved ones getting sick and/or dying	140 (72.9)	134 (69.8)	274 (71.4)	0.46	0.29
Feeling socially isolated/lonely	100 (52.1)	72 (37.5)	172 (44.8)	8.26	0.002
Feeling like I can’t share my concerns/feelings safely with others	73 (38.0)	63 (32.8)	136 (35.4)	1.14	0.17
Difficulty sleeping due to increased stress from the pandemic	67 (34.9)	73 (38.0)	140 (36.5)	0.40	0.30
Difficulty making arrangements for dependent care (e.g., children, elderly relatives)	32 (16.7)	19 (9.9)	51 (13.3)	3.82	0.04
Uncertainty about how long the pandemic will continue	165 (85.9)	169 (88.0)	334 (87.0)	0.37	0.32
Fears of societal instability	133 (69.3)	124 (64.6)	257 (66.9)	0.95	0.19
Personal resource worries (e.g., financial concerns, not having a will/advance directive in place, not having adequate supplies at home)	83 (43.2)	114 (59.4)	197 (51.3)	10.02	0.001
Total Responses	**192**	**192**	**384**		

**Table 3 ijerph-20-05953-t003:** Odds ratio estimates of professional category as a predictor of moral distress outcomes in locked-out versus locked-in HCW, controlling for baseline demographics and years of experience.

Professional Category	Increasing Baseline Moral Distress			High Baseline Moral Distress		
	Odds Ratio	95% Confidence Interval	*p*-Value	Odds Ratio	95% Confidence Interval	*p*-Value
**Allied Health**	0.26	0.03, 2.63	<0.001	0.64	0.03, 8.25	0.98
**Massage Therapy**	1.51	0.78, 2.93	0.54	1.04	0.47, 2.32	0.22
**Medicine**	0.31	0.10, 0.96	<0.001	0.14	0.02,	0.23
**Mental Health**	1.76	0.26, 11.81	0.52	0.74	0.07, 7.42	0.98
**Nursing**	1.31	0.64, 2.71	<0.001	1.17	0.51, 2.71	0.39
**Dual Profession**	1.03	0.44, 2.40	<0.001	0.91	0.33, 2.50	0.59
**Other**	0.78	0.41, 1.49	0.01	0.81	0.31, 2.09	0.50
**All Professions**	0.72	0.52, 0.98	0.37	0.81	0.06, 1.05	0.04

**Table 4 ijerph-20-05953-t004:** Odds ratio estimates of locked-out versus locked-in HCW on moral distress, burnout, and well-being, controlling for baseline demographics and years of experience.

Scale	Outcomes	OR	95% CI	*p*-Value
Ordinal	Moral Distress	0.55	0.38, 0.81	0.30
	WBI	1.19	0.82, 1.72	0.22
	Mini-Z	1.57	1.06, 2.33	0.20
High Scores	Moral Distress	0.46	0.28, 0.73	0.02
	WBI	1.34	0.81, 2.22	0.19
	Mini-Z	1.38	0.68, 2.92	0.16

## Data Availability

Data from this project is not publicly available due to the potential of identifiability of video blog recordings.

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
