# Peer review of "“I Don’t Want to Go to Work”: A Mixed-Methods Analysis of Healthcare Worker Experiences from the Front- and Side-Lines of COVID-19"

_ijerph, 2023, doi:10.3390/ijerph20115953_

Round 1
Reviewer 1 Report
The study is interesting, but some suggestions for improvement are given below:
There are some technical mistakes. Please, follow the instru tions for authors.
Please, add the comparison with other research in the Discussion.
Please, add the limitations of the study in the Conclusions, not Discussion. Add also sugesstions for further research.
Author Response
Thank you for your helpful comments and recommendations. We have made the following changes:
-
We have converted our submission to the submission template and updated our format.
-
Comparison to other research added to the discussion.
-
We have moved our limitations statement to the conclusions section and added suggestions for further research.
We feel these changes have greatly strengthened our submission, and we look forward to your review.
Reviewer 2 Report
The study has a lot of potential and is interesting. The literature review should be substantially improved and expanded, as well as the reasons why it is innovative (at the begining and at the end of the paper). It is suggested that you review the presentation of the objective of the study and the common thread. Moreover authors should improve the presentation of the qualitative contributions together with the quantitative ones, achieving an logical integration.Graphs or tables should be added to show the reader these qualitative as well as quantitative results.
Author Response
Thank you for this helpful review. We have heavily modified our manuscript and transferred to the submission template to provide better clarity and incorporation of graphs and figures. We have also significantly expanded our literature review.
Reviewer 3 Report
Thank you for the opportunity in reviewing the article "I don't want to go to work": a mixed methods analysis of the experiences of front and side line health workers in covid-19.
Very interesting, current and pertinent topic.
Taking into account the different sections, I suggest the following changes:
Abstract
Can you please revise the results, particularly the qualitative analysis, as the information is not clear.
The key words should not be repeated with the words of the title.
Introduction
Can you please clarify and better substantiate the relevance of the study?
Methodology
Please further clarify the type of study, mixed, qualitative or quantitative? If it is qualitative, and quantitative, why not separate and conduct two studies? What is the relevance?
I did not verify information on ethical issues, were they not provided for?
Results
Check table 2, and clarify why the N is in the row of the variable.
Check table 3, and clarify the column of Estimate, what is the purpose of this column?
Discussion
They should further discuss the differences regarding work-related stressors and look for possible explanations for them. And even regarding all the variability found in the study.
Author Response
Thank you for this helpful review. In response to your comments, we have made the following changes:
- Results revised in abstract to clarify the analysis.
- We have removed key words which were also included in the title.
- Additional literature cited to clarify the study’s relevance
- We have restructured our methods section to better demonstrate the mixed-methods analysis conducted.
- We have added a statement on ethical issues.
- Tables corrected and clarified in text. Thank you
- Discussion expanded to cover work-related stressors and variability.
Reviewer 4 Report
The article deals with an important topic. The substantive description of the research premises is very valuable.
However, the Authors should enrich the literature review. Currently, a lot of articles have been published on Covid-19 and professional burnout. These articles deal with the theory of the psychology of occupational burnout and its relevance to occupational burnout in the era of Covid-19. Many studies are based on primary research conducted among professional and socio-demographic groups examining their responses to the pandemic and lockdown. The Authors of the reviewed article limited themselves to only a few items. For example, the Authors can use:
- Kwiatkowska-Ciotucha, D., ZaÅ‚uska, U., Åšlazyk-Sobol, M., Lehesvuori, M. & Polak, A. (2019). Occupational Burnout in Health Care – Analysis of Systemic and Organisational Risks as Well as Possible Preventive Actions. Econometrics, 23(4) 43-62. https://doi.org/10.15611/eada.2019.4.04
- Zaluska, U., Kwiatkowska-Ciotucha, D. & Slazyk-Sobol, M. (2020). Burnout Syndrome as the Example of Psychological Costs of Work – Empirical Studies among Human-Oriented Professions in Poland. IBIMA Business Review, Vol. 2020 (2020), Article ID 430264, DOI: 10.5171/2020.430264
- ZaÅ‚uska,U., Åšlazyk-Sobol, M. & Kwiatkowska-Ciotucha, D. (2018). Burnout and Its Correlates − An Empirical Study Conducted Among Education, Higher Education and Health Care Professionals. Econometrics, 22(1) 26-38. https://doi.org/10.15611/eada.2018.1.02
Considering the article's number of authors, an additional literature review should not be time-consuming.
My second objection concerns the analysis carried out. The authors in supplementary materials include many tables with results and additional descriptions. However, the article does not specify which table or appendix of supplementary materials the Authors use.
The authors fell into the trap of very general conclusions. When presenting any table, it is necessary to carefully refer to its content and precisely provide the table number in which the discussed values are located. Without this, the correctness of the inference cannot be assessed.
Please note that statements such as „Current research into the burden of the COVID-19 pandemic on HCW is often limited to single perspective/site narratives” (lines 43-44) should be supported by a reference to literature or your own research.
In the abstract, the Authors did not indicate the purpose of the study.
Author Response
Thank you for this helpful feedback. We have expanded our literature review. Thank you for your article recommendations. Two of these articles were included in our updated literature review. We have clarified in text references to tables and appendix materials and removed extraneous supplementary materials.
We have revised our statement about limitations of current COVID-19 burnout literature and added citations.
Finally, we have added a purpose statement to the abstract.
We feel these changes have significantly improved our manuscript. Thank you for your review.
Round 2
Reviewer 3 Report
Thank you for the new opportunity to review the paper.
Sorry but I can't see where the ethical information you have added is? Which line?
Author Response
Hello,
Thank you for your review. The IRB review is mentioned in line 82 of the manuscript and repeated in the IRB statement on line 474. We would be happy to add any additional language to these statements and welcome your recommendations.
Reviewer 4 Report
I am satisfied with the changes made by the authors.
Author Response
Thank you for your review!